# Creatinine clearance is key to solving the enigma of sex difference in in-hospital mortality after STEMI: Propensity score matching and mediation analysis

**Parisa Janjani[1], Nahid Salehi[1], Mohammad Rouzbahani[1], Soraya Siabani[1,2], Meysam Olfatifar** [1,3]*

1 Cardiovascular Research Center, Health Institute, Imam-Ali hospital, Kermanshah University of Medical Sciences, Kermanshah, Iran, 2 Department of Health Education and Health Promotion, Kermanshah University of Medical Sciences, Kermanshah, Iran, 3 Gastroenterology and Hepatology Diseases Research Center, Qom University of Medical Sciences, Qom, Iran

* ol.meysam92@gmail.com

**Data Availability Statement:** All relevant data are within the manuscript and its Supporting information files.

## Abstract

### Background

The precise impact of sex difference on in-hospital mortality in ST-elevation myocardial infarction (STEMI) patients are unclear, and the studies are no longer consistent. Therefore, we sought to evaluate the impact of sex differences in a cohort of STEMI patients.

### Methods

We analyzed the data of 2647 STEMI patients enrolled in the Kermanshah STEMI Cohort from July 2017 to May 2020. To accurately clarify the relationship between sex and hospital mortality, propensity score matching (PSM) and causal mediation analysis was applied to the selected confounder and identified intermediate variables, respectively.

### Results

Before matching, the two groups differed on almost every baseline variable and in-hospital death. After matching with 30 selected variables, 574 male and female matched pairs were significantly different only for five baseline variables and women were no longer at greater risk of in-hospital mortality (10.63% vs. 9.76%, p = 0.626). Among the suspected mediating variables, creatinine clearance (CLCR) alone accounts for 74% (0.665/0.895) of the total effect equal to 0.895(95% CI: 0.464–1.332). In this milieu, the relationship between sex and in-hospital death was no longer significant and reversed -0.233(95% CI: -0.623–0.068), which shows the full mediating role of CLCR.

### Conclusion

Our research could help address sex disparities in STEMI mortality and provide a consequence. Moreover, CLCR alone can fully explain this relationship, which can highlight the

**Funding:** This work was supported by the Cardiovascular Research Center, Institute of Health, Kermanshah University of Medical Sciences (Grant #95449). PJ received this grant.

**Competing interests:** The authors have no conflicts of interest.

importance of CLCR in predicting the short-term outcomes of STEMI patients and provide a useful indicator for clinicians.

## Introduction

Over the years, many studies have assessed differences in mortality following STEMI between men and women [1–3]. However, the results obtained in this area are inconsistent, independently of the differences between the studies in terms of study type, study location and sample size. Although many studies have proven the role of the female sex in this field, others have disputed distinctions between the two groups [1–4]. The kind and number of variables included in statistical models employed in research, mainly regression models, may attract our attention at first glance, which is acceptable since the research on this subject is not totally or even partially consistent in terms of the factors evaluated.

Is it the kind and number of factors that have generated this heterogeneity? It is impossible to give health professionals and specialists a solid clinical standard in this field. This study's scope does not allow for a clear response to this issue. However, aside from the relevance of clinical information in variable selection, the statistical approaches utilized in this field are quite effective in both variable selection and analysis. Depending on the type of outcomes, the majority of the available data is based on observational research. As a result, designing clinical trials in this area is difficult to eliminate potential and hidden confounding factors between the two groups [5, 6].

One of the solutions advocated by researchers and epidemiologists in this field is to employ propensity score matching (PSM) methods since they can help us explain the impact of confounding factors more precisely, which standard regression methods cannot achieve[5, 6]. It should be emphasized that the findings produced in these models, in turn, are in the group of variables chosen for study, and a lack of suitable variable selection might lead to biased results [7–9]. In other words, only confounder and not intermediate variables should be included in the PSM analysis to reduce bias [10]. An intermediate variable is a variable that interacts between exposure/ treatment and outcome causal chain. In this context, causal mediation analysis is an acceptable solution, but these methods are often ignored by researchers, not only in studies using PSM analysis but also in other studies [11].

Taken together, despite the caveats and recommendations mentioned in the evidence, many studies utilizing PSM to explain the variations in mortality between men and women in STEMI patients have not followed or addressed these guidelines. Hence, we attempt to correctly describe this issue by selecting and identifying the counfander and intermediate varibles respectively. As a result, we aimed to avoid using mediator variables in the model as much as feasible, hence minimizing the model's final bias [9]. It is hoped that the findings of this study will pave the way for sound clinical decisions.

## Methods

### Study population

The current study is based on a cohort of STEMI patients from the Imam Ali Hospital at the Kermanshah University of Medical Sciences. From July 2017 to May 2020, we included data for all STEMI patients enrolled in the cohort, totaling 2,816 patients. Since there have been several studies on the characteristics of the cohort, we will only briefly mention it. Adults 18 years of age and older with angina or other comparable symptoms lasting more than 20 minutes in

the previous 24 hours before admission, as well as the presence of left bundle branch block (LBBB) and ST-segment elevation in the diagnostic elektrokardiogramm (EKG), are classified as having suspected STEMI in this cohort. STEMI was first diagnosed by the emergency doctor and then confirmed by the study quality control doctor. Patients hospitalized for another reason and then developed STEMI and patients who developed STEMI while undergoing percutaneous coronary angiography (PCI) or bypass surgery were excluded from the cohort. Also, patients admitted to another hospital 24 hours before admission to Imam Ali Hospital were excluded from the study. Our research has been ethically approved by the Ethics Committee at Kermanshah University of Medical Sciences, Kermanshah, Iran (NO. KUMS.REC.1395.252) and is in accordance with the principles presented in the Declaration of Helsinki. All participants in this study provided written informed consent (patients without written informed consent were not included in the study). Nevertheless, in the case of extreme illness or death in the hospital, the patient's relatives signed the informed consent form.

## Variable selection

First of all, we attempted to impute missing data for study variables using a multiple imputation (MI) approach. According to an interesting study [12], we did not use the proportion of missing data for each variable to decide whether to remove it from the analysis. Instead, we used the fraction of missing information (FMI) as a selection criterion (the FMI for the variables examined and other information on this are reported in S2 File). On the other hand, using all of the obtained variables for propensity score matching (PSM) analysis is not required based on causal inference principles [7, 13]. In other words, the variable under consideration must be a correct confounder or not be in the causal path between the exposure and the outcome (mediating variable) [8]. Hence, we attempted to identify the most relevant variables by reviewing the literature [1, 14–17] and considering the experts' opinions. In such a way, we finally included 30 variables out of 55 variables in the PSM model (Tables 1 and 2) based on outcome adaptive lasso. Likewise, the creatinine clearance (CLCR) and early hemoglobin (Hb) as complete mediators and the lowest hemoglobin (Hb), PIC, and diabetes status variables as partial mediators were not included in the model, and their effect was estimated separately. The details of the selection of variables are given in S2 File. In addition, the method of calculating CLCR is also mentioned in S2 File.

## Outcome

In-hospital death was the outcome of our study. As a result, we only consider the data of 2,647 persons in our study out of 2,816 patients whose data were collected from July 2017 to May 2020. Because 32 persons went lost during the follow-up, 137 people's information was linked to follow-up death. These 137 people were alive until the moment they were discharged from the hospital, and in the follow-up, death was imminent for them. Out of 172 in-hospital deaths, only ten were non-cardiovascular, and 162 were cardiovascular diseases.

## Statistical analysis

The t-test or Mann-Whitney test was applied in terms of normality for quantitative variables, and the chi-square test was utilized before matching for qualitative variables to compare the discussed variables between males and females [18]. Similarly, following matching, the paired t-test or Wilcoxon rank-sum test was used to compare quantitative variables, and the chi-square test was used to evaluate qualitative factors between the two groups.

**Propensity score matching.** We utilized the propensity score matching (PSM) technique in this study to balance the distribution of variables between men (as the control) and women

**Table 1. Comparison of basic and admission covariates before and after propensity score matching (PSM) based on sex.**

| Variables | Sex (before PSM) | | P | Sex (after PSM) | | P |
|---|---|---|---|---|---|---|
| | Male(n = 2073) | Female(n = 574) | | Male(n = 574) | Female(n = 574) | |
| **Inclusion and exclusion related and basic variables** | | | | | | |
| Age (year) | 58.83±11.59 | 65.34±11.34 | 0.001 | 64.14±11.75 | 65.34±11.34 | 0.068 |
| BMI (kg/m$^2$) | 26.17±4.02 | 26.91±4.43 | 0.001 | 26.55±4.09 | 26.91±4.43 | 0.141 |
| HTN | | | 0.001 | | | 0.001 |
| No | 1333(64.33) | 178(31.01) | | 237(41.29) | 178(31.01) | |
| Yes | 719(34.70) | 391(68.12) | | 327(56.97) | 391(68.12) | |
| Unknown | 20(0.97) | 5(0.87) | | 10(1.74) | 5(0.87) | |
| Old MI | | | 0.011 | | | 0.043 |
| No | 1736(83.74) | 504(87.80) | | 502(87.46) | 504(87.80) | |
| Yes | 268(12.93) | 48(8.36) | | 62(10.80) | 48(8.36) | |
| Unknown | 69(3.33) | 22(3.83) | | 10(1.74) | 22(3.83) | |
| CCS class | | | 0.418 | | | 0.248 |
| I | 1140(54.99) | 306(53.31) | | 302(52.61) | 306(53.31) | |
| II | 897(43.27) | 258(44.95) | | 259(45.12) | 285(44.95) | |
| III | 17(0.82) | 3(0.52) | | 8(1.39) | 3(0.52) | |
| IV | 3(0.14) | 3(0.52) | | 0(0.00) | 4(0.70) | |
| Unknown | 16(0.77) | 4(0.70) | | 5(0.87) | | |
| CHF | | | 0.001 | | | 0.510 |
| No | 1954(94.26) | 511(89.02) | | 522(90.94) | 511(89.02) | |
| Yes | 52(2.51) | 29(5.05) | | 26(4.53) | 29(5.05) | |
| Unknown | 67(3.23) | 34(5.92) | | 26(4.53) | 34(5.92) | |
| HLP | | | 0.001 | | | 0.001 |
| No | 1605(77.42) | 323(56.27) | | 370(64.46) | 323(56.27) | |
| Yes | 395(19.05) | 232(40.42) | | 164(28.57) | 232(40.42) | |
| Unknown | 73(3.52) | 19(3.31) | | 40(6.97) | 19(3.31) | |
| Previous PVD | | | 0.026 | | | 0.418 |
| No | 2044(98.60) | 556(96.86) | | 563(98.08) | 556(96.86) | |
| Yes | 6(0.29) | 3(0.52) | | 2(0.35) | 3(0.52) | |
| Unknown | 23(1.11) | 15(2.61) | | 9(1.57) | 15(2.61) | |
| Previous AF | | | 0.014 | | | 0.444 |
| No | 1913(92.28) | 509(88.68) | | 517(90.07) | 509(88.68) | |
| Yes | 2(0.10) | 0(0.00) | | 57(9.93) | 65(11.32) | |
| Unknown | 158(7.62) | 65(11.32) | | - | - | - |
| Previous CABG | | | 0.678 | | | 0.444 |
| No | 2002(96.58) | 558(97.21) | | 558(97.21) | 558(97.21) | |
| Yes | 70(3.38) | 16(2.79) | | 15(2.61) | 16(2.79) | |
| Unknown | 1(0.05) | 0(0.00) | | 1(0.17) | 0(0.00) | |
| Previous PCI | | | 0.063 | | | 0.542 |
| No | 1929(93.05) | 547(95.30) | | 541(94.25) | 547(95.30) | |
| Yes | 141(6.80) | 25(4.36) | | 32(5.57) | 25(4.36) | |
| Unknown | 3(0.14) | 2(0.35) | | 1(0.17) | 2(0.35) | |
| Previous stroke | | | 0.082 | | | 0.246 |
| No | 1975(95.27) | 535(93.21) | | 530(92.33) | 535(93.21) | |
| Yes | 88(4.25) | 37(6.45) | | 37(6.45) | 37(6.45) | |
| Unknown | 10(0.48) | 2(0.35) | | 7(1.22) | 2(0.35) | |
| Current smoking | | | 0.001 | | | 0.115 |

(*Continued*)

**Table 1.** (Continued)

| Variables | Sex (before PSM) | | P | Sex (after PSM) | | P |
|---|---|---|---|---|---|---|
| | Male(n = 2073) | Female(n = 574) | | Male(n = 574) | Female(n = 574) | |
| No | 840(40.52) | 499(86.93) | | 478(82.40) | 499(86.93) | |
| Yes | 1226(59.14) | 71(12.37) | | 94(17.60) | 71(12.37) | |
| Unknown | 7(0.34) | 4(0.70) | | 2(0.35) | 4(0.70) | |
| Dialysis | | | 0.615 | | | 1.00 |
| No | 2069(99.81) | 572(99.74) | | 572(99.65) | 572(99.65) | |
| Yes | 3(0.14) | 1(0.17) | | 1(0.17) | 1(0.17) | |
| Unknown | 1(0.05) | 1(0.17) | | 1(0.17) | 1(0.17) | |

BMI: body mass index, HTN: hypertension, MI: myocardial infarction, CCS: canadian cardiovascular society, CHF: chronic heart failure, HLP: hyperlipidemia, PCI: percutaneous coronary intervention, PVD: peripheral vascular disease, AF: atrial fibrillation, CABG: coronary artery bypass grafting

(as the case). We used efficient research by Zhao et al. for this aim to execute the model accurately and optimally [9]. In this way, we estimated propensity scores: using the generalized linear model (GLM), applying the optimal matching technique (in this approach, there is no need to determine the caliber matching value) and with a case-to-control ratio of 1:1. The balance or unbalance of the studied variables was next evaluated, and the effect of hidden confounders was ultimately considered using sensitivity analysis. We utilized Rosenbaum's Sensitivity Analysis approach for this purpose. All statistical analyzes were performed in R software version 4.1.2.

**Mediating analysis.** In the medical sciences, describing causal pathways is essential to precisely estimate the causal effects. As a result, we explored the causal relationship between sex, in-hospital mortality, and suspected mediating variables, as depicted in Fig 1.

Hence, we estimated the direct effects (the total effect of sex on the outcome without considering the mediator variable) and indirect effects (the effect of sex on the outcome, considering the effect of the mediator variable) of sex [19]. We used the method presented in the work of Qingzhao et al. [20], which is provided in the mma package of R software.

## Results

### Basic and admission information

Overall, 21.69% of the subjects were female. The mean and standard deviation of age in women was 65.34 ±11.34 and 58.83 ±11.95 in men, which shows that the women in our study are older. Tables 1 and 2 showed the other characteristics of the subjects before and after using PSM.

Before the matching, in terms of basic and admission covariates, only 5 of the 14 variables examined were not significantly different between the two groups. The females were shown to have a relatively higher BMI, higher rate of hypertension (HTN) (68.12% vs. 34.70%, p = 0.001), congestive heart failure (CHF) (5.05% vs. 2.51%, p = 0.001), hyperlipidemia (HLP), and Peripheral vascular disease (PVD) than man (Table 1). However, regarding the history of myocardial infarction (MI) (12.93% vs. 8.36%, p = 0.011) and smoking (59.14% VS. 12.37%, p = 0.001), men snatched the lead from women (Table 1). After matching, the significant variable in this category was HLP, HTN, and old MI, that HTN being more in women (68.12% VS. 56.97%, p0.001) and MI in men (10.80 vs. 8.36, p = 0.043) (Table 1). The characteristics of other variables not included in the analysis are shown in S2, S3 Tables in S2 File.

**Table 2. Comparison of initial assessment, coronary anatomy and PCI procedure and hospitalization variables before and after propensity score matching (PSM) based on sex.**

| Variables | Sex (before PSM) | | P | Sex (after PSM) | | P |
|---|---|---|---|---|---|---|
| | Male(n = 2073) | Female(n = 574) | | Male(n = 574) | Female(n = 574) | |
| **Initial assessment, coronary anatomy and PCI procedure** | | | | | | |
| TIMI flow before dilatation | | | 0.087 | | | 0.556 |
| 0 | 1449(69.90) | 372(64.81) | | 370(64.46) | 372(64.81) | |
| 1 | 422(20.36) | 138(24.04) | | 133(23.17) | 138(24.04) | |
| 2 | 166(8.01) | 47(8.19) | | 60(10.45) | 47(8.19) | |
| 3 | 7(0.34) | 4(0.70) | | 3(0.52) | 4(0.70) | |
| Unknown | 29(1.40) | 13(2.26) | | 8(1.39) | 13(2.26) | |
| TIMI flow after dilatation | | | 0.092 | | | 0.641 |
| 0 | 18(0.87) | 9(1.57) | | 11(1.92) | 9(1.57) | |
| 1 | 12(0.58) | 2(0.35) | | 6(1.05) | 2(0.35) | |
| 2 | 76(3.67) | 31(5.40) | | 27(4.70) | 31(5.40) | |
| 3 | 1936(93.39) | 519(90.42) | | 516(89.90) | 519(90.42) | |
| Unknown | 31(1.50) | 13(2.26) | | 14(2.44) | 13(2.26) | |
| Thrombectomy duration PCI | | | 0.001 | | | 0.469 |
| No | 1512(72.94) | 458(79.79) | | 448(78.05) | 458(79.79) | |
| Yes | 561(27.06) | 116(20.21) | | 126(21.95) | 116(20.21) | |
| Stent | | | 0.667 | | | 0.197 |
| No | 370(17.85) | 98(17.07) | | 115(20.03) | 98(17.07) | |
| Yes | 1703(82.15) | 476(82.93) | | 459(79.97) | 476(82.93) | |
| Number of epicardial territories with stenosis > 50% | | | 0.029 | | | 0.731 |
| 0 | 17(0.82) | 5(0.87) | | 3(0.52) | 5(0.87) | |
| 1 | 607(29.28) | 144(25.09) | | 138(24.04) | 144(25.09) | |
| 2 | 698(33.67) | 183(31.88) | | 195(33.97) | 183(31.88) | |
| 3 | 751(36.23) | 241(41.99) | | 238(41.46) | 241(41.99) | |
| Unknown | 0(0.00) | 1(0.17) | | 0(0.00) | 1(0.17) | |
| left main stem (LMS) stenosis>50% or fractional flow reserve (FFR) less than 0.8 | | | 0.886 | | | 0.861 |
| No | 2021(97.49) | 559(97.39) | | 556(96.86) | 559(97.39) | |
| Yes, unprotected | 50(2.41) | 14(2.44) | | 17(2.96) | 14(2.44) | |
| Yes, unprotected by CABG | 2(0.10) | 1(0.17) | | 1(0.17) | 1(0.17) | |
| AF on qualifying EKG | | | 0.417 | | | 0.415 |
| No | 2024(97.64) | 557(97.04) | | 552(96.17) | 557(97.04) | |
| Yes | 49(2.36) | 17(2.96) | | 22(3.83) | 17(2.96) | |
| KLLIP class | | | 0.471 | | | 0.434 |
| Class I | 1851(89.29) | 518(90.24) | | 502(87.46) | 518(90.24) | |
| Class II | 94(4.53) | 21(3.66) | | 31(5.40) | 21(3.66) | |
| Class III | 12(0.58) | 6(1.05) | | 6(1.05) | 6(1.05) | |
| Class IV | 116(5.60) | 29(5.05) | | 35(6.10) | 29(5.05) | |
| **During hospitalization until discharge** | | | | | | |
| ESR (mm/h) | 10.18±9.69 | 16.47±14.79 | 0.001 | 13.64±13.32 | 16.47±14.79 | 0.001 |
| HDL (mg/dL) | 40.54±8.67 | 44.03±10.31 | 0.001 | 42.79±8.90 | 44.03±10.31 | 0.024 |
| Early Cr (mg/dL) | 1.16±0.325 | 1.08±0.290 | 0.001 | 1.13±0.234 | 1.08±0.290 | 0.001 |
| AF in hospital | | | 0.021 | | | 0.815 |
| No | 1985(95.75) | 536(93.38) | | 534(93.03) | 536(93.38) | |
| Yes | 88(4.25) | 38(6.62) | | 40(6.97) | 38(6.62) | |
| HF in hospital | | | 0.513 | | | 0.178 |

*(Continued)*

**Table 2.** (Continued)

| Variables | Sex (before PSM) | | P | Sex (after PSM) | | P |
|---|---|---|---|---|---|---|
| | Male(n = 2073) | Female(n = 574) | | Male(n = 574) | Female(n = 574) | |
| No | 1598(77.09) | 435(75.78) | | 415(72.30) | 435(75.78) | |
| Yes | 475(22.91) | 139(24.42) | | 159(27.70) | 139(24.22) | |
| Transfusion | | | 0.055 | | | 0691 |
| No | 1915(92.38) | 443(89.90) | | 520(90.59) | 516(89.90) | |
| Yes | 158(7.62) | 11(10.10) | | 54(9.41) | 58(10.10) | |
| Staged PCI | | | 0.083 | | | 0.158 |
| No | 1558(75.16) | 443(77.18) | | 441(76.83) | 443(77.18) | |
| Performed during the index hospitalization | 49(2.36) | 11(1.92) | | 12(2.09) | 11(1.92) | |
| Planned but canceled because of patient refusal | 4(0.19) | 5(0.87) | | 0(0.00) | 5(0.87) | |
| Planned to be performed on subsequent admission | 460(22.19) | 115(20.03) | | 121(21.08) | 115(20.03) | |
| Unknown | 2(0.10) | 0(0.00) | | - | - | |
| Worst KLLIP class | | | 0.009 | | | 0.070 |
| Class I | 1508(72.74) | 406(70.73) | | 374(65.16) | 406(70.73) | |
| Class II | 305(14.71) | 70(12.20) | | 97(16.90) | 70(12.20) | |
| Class III | 32(1.54) | 18(3.14) | | 13(2.26) | 18(3.14) | |
| Class IV | 228(11.00) | 80(13.94) | | 90(15.68) | 80(13.94) | |
| Death | | | 0.001 | | | 0.626 |
| No | 1962(94.65) | 513(89.37) | | 518(90.24) | 513(89.37) | |
| Yes | 111(5.35) | 61(10.63) | | 56(9.76) | 61(10.63) | |

TIMI: Thrombolysis in Myocardial Infarction, PCI: percutaneous coronary intervention, CABG: coronary artery bypass grafting, AF: atrial fibrillation, EKG: electrocardiogram, ESR: erythrocyte sedimentation rate, HDL: high-density lipoprotein, Cr: creatinine, HF: heart failure

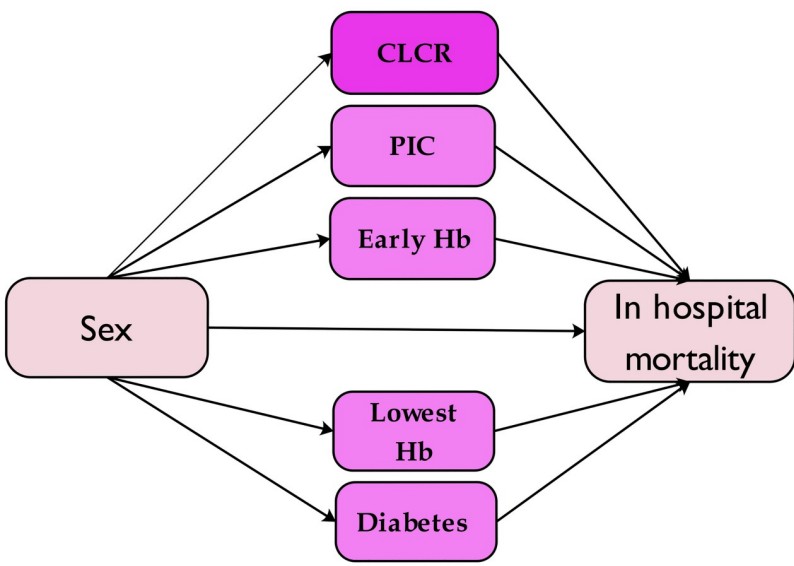

**Fig 1. Conceptual model of mediating effects of creatinine clearance (CLCR), percutaneous coronary intervention (PCI), diabetes, early hemoglobin (Hb) and lowest hemoglobin (Hb) variables with gender (independent variable) and in hospital mortality(outcome).**

### Initial assessment, PCI procedure, and hospitalization

Among the remaining 16 variables related to initial assessment, PCI procedure, and hospitalization, 7 variables had significant differences between the two groups before matching (Table 2).

Women experienced higher mean erythrocyte sedimentation rate (ESR) (16.47 vs. 10.18, p = 0.001) and high-density lipoprotein (HDL) (44.03 vs. 40.54, p = 0.001) than men. Also, in terms of in-hospital atrial fibrillation (AF) (6.62 vs. 4.25, p = 0.021) and Worst KLLIP class, higher values were observed in women (Table 2). Although, thrombectomy was more common in men than women (27.06% vs. 20.21%, p = 0.001) the early creatinine (Cr) was higher in men than women (Table 2). After matching, ESR, HDL, and early Cr had significant differences between the two groups with the same previous pattern (Table 2). Finally, females experienced higher in-hospital mortality than men before matching (10.63% vs. 5.35%, p = 0.001) (Table 2). After using PSM, the relationship between gender and death was no longer significant, although a higher percentage of women experienced death (10.63% vs. 9.76%, p = 0.626) (Table 2). The characteristics of other variables not included in the analysis are shown in S2, S3 Tables in S2 File.

### Propensity score matching

In our study, PSM with the optimal technique resulted in 574 matched male-female pairs that were balanced for most of the studied variables (Fig 2 and Table 2 and S2, S4 Tables in S2 File).

Hence, based on the standardized mean difference (SMD), the distribution of 5 of the 30 variables in the PSM analysis was not balanced between the two groups (S2, S4 Tables in S2 File). Additionally, one of the studied variables was not balanced between the groups regarding variance ratio (VR) S2, S4 Tables in S2 File.

**Sensitivity analysis.** Matching estimates are valid whenever hidden confounders do not influence them; therefore, we used sensitivity analysis to assess the impact of these variables. The findings of Rosenbaum Sensitivity Analysis revealed that unobservable confounders slightly influenced the differential assignment of patients, whereas 0.1 increments of the quantity of gamma (differential likelihood of being allocated to the treatment group owing to hidden confounders) increased the p-value from 0.162 to 0.308. In other words, hidden confounders have no impact in discriminating between two outcome classes [21].

### Mediation analysis

As mentioned before, due to the importance of mediation analysis, in this section we tried to estimate the direct effects of gender as well as its indirect effects through mediating variables. As shown in Fig 3 and Table 3, the total effect (sum of direct and indirect effects) was equal to 0.895 (95% CI: 0.464–1.332) which is significant (P = 0.001).

Likewise, the indirect effect or Average Causal Mediation Effect (ACME) related to the CLCR variable had the greatest impact on the relationship between gender and in-hospital mortality 0.665 (95% CI: 0.468–0.885).

In other words, CLCR alone can explain 74% (0.665/0.895) of the sexual difference in in-hospital mortality. Afterward, the PIC variable had a positive and significant effect on this relationship and covered 24% (0.212/0.895) of the relationship between sex and mortality of STEMI patients. However, the effect of diabetes, early Hb, and lowest Hb variables were insignificant (Table 2). Similarly, the sex effect or the average direct effect (ADE) not remained significant and reversed from 0.742(95% CI: 0.415–1.069) to -0.233(95% CI: -0.623–0.068). The results of this section are extremely clinically important because they show that CLCR can greatly explain the relationship between sex and in-hospital mortality in STEMI patients.

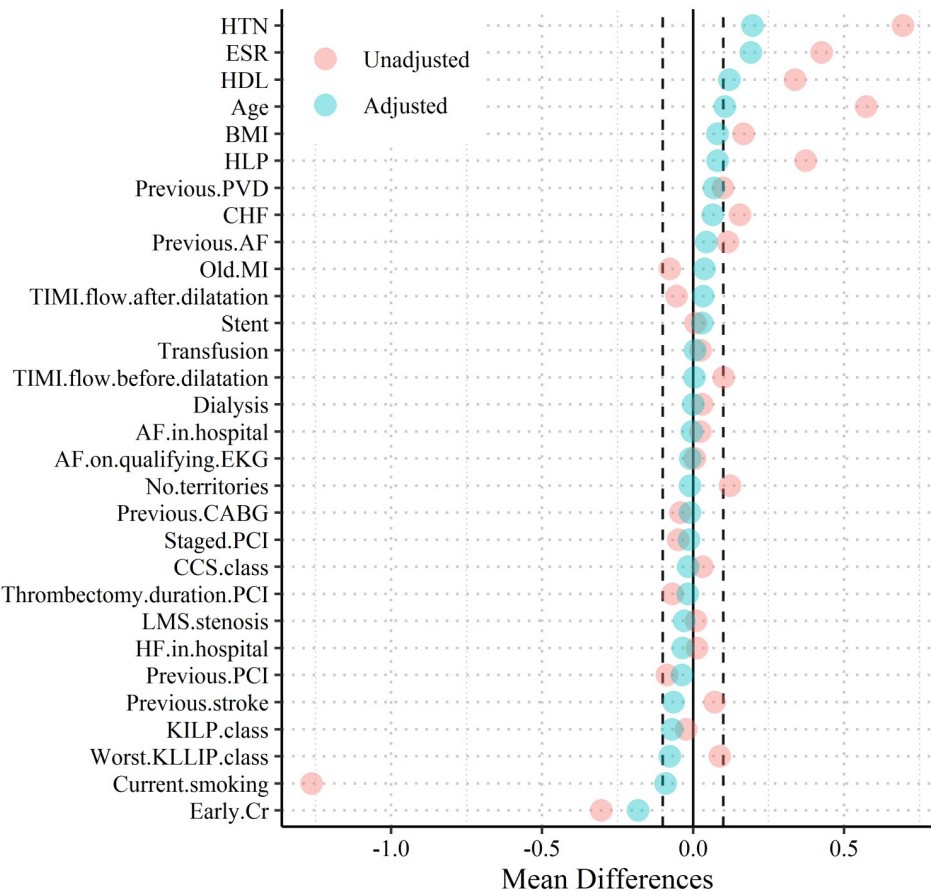

**Fig 2. Love plot of selected covariates, after (blue circles) and before (red circles) PSM analysis by sex, the very narrow width of the dashed lines around the blue points suggests that the variables have been well-balanced after matching.** HTN hepertention, ESR erythrocyte sedimentation rate, HDL high-density lipoprotein, BMI body mass index, HLP hyperlipidemia, PVD peripheral vascular disease, CHF chronic heart failure, AF atrial fibrillation, MI myocardial infarction, TIMI Thrombolysis in Myocardial Infarction, EKG electrocardiogram, CABG coronary artery bypass grafting, CCS canadian cardiovascular society, PCI percutaneous coronary intervention, LMS left main stem, HF heart failure, Cr creatinine.

The characteristics of mediating variables between men and women are shown in S2 Table in S2 File.

## Discussion

Overall, our work assisted in elucidating the causal association between sex and in-hospital mortality in STEMI patients by accurately identifying the variables needed for PSM analysis and by doing a mediating analysis. In other words, this study provided a decent step in explaining the clinical care of STEMI patients based on sex differences. Our results revealed that the difference between men and women was no longer significant after the PSM application. However, the differences persisted in some investigated variables, such as HLP, HTN, old MI, ESR, HDL, and early Cr. These results suggest that the other 25 variables may well neutralize the independent role of gender in in-hospital mortality Fig 2 and Tables 1 and 2. Similarly, as an essential clinical insight, our study revealed solid evidence of the CLCR variable's mediating role in the causal link between sex and in-hospital death in STEMI patients (Figs 1 and 3).

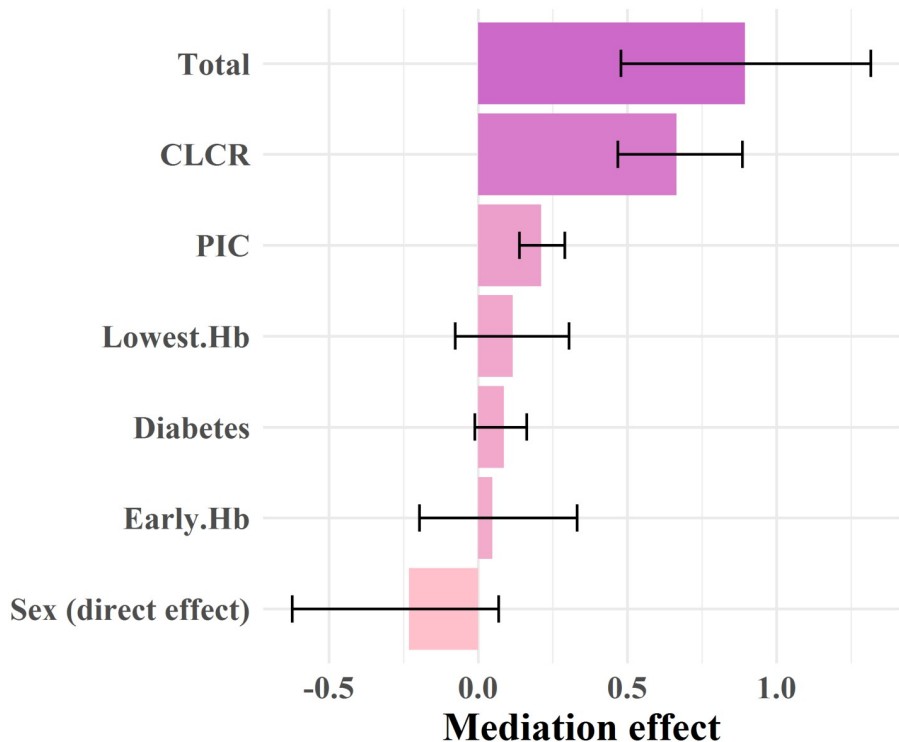

**Fig 3. Mediating effect of creatinine clearance (CLCR), percutaneous coronary intervention (PCI), diabetes, early hemoglobin (Hb) and lowest hemoglobin (Hb) variables.**

Thus, the indirect effect of CLCR accounts for about 74% of the contribution of sex in this causal connection. This finding can create a revolution in the clinical management of patients. Therefore, it is hoped that more women's in-hospital deaths will be controlled by proper management of CLCR.

**Table 3. Relationship between gender and in-hospital mortality in STEMI patients in simple logistic model and multiple mediation model.**

|  | Effect* | Mean | SD | 95% CI |
|---|---|---|---|---|
| Simple model |  |  |  |  |
| Sex | 0.742 | - | - | 0.415–1.069 |
| Mediating model |  |  |  |  |
| Total effect | 0.895 | 0.898 | 0.221 | 0.479–1.316 |
| Sex (direct effect) | -0.232 | -0.225 | 0.192 | -0.623–0.068 |
| Indirect effects |  |  |  |  |
| CLCR (mL/min) | 0.665 | 0.673 | 0.110 | 0.468–0.885 |
| PIC | 0.212 | 0.201 | 0.039 | 0.137–0.290 |
| Diabetes | 0.086 | 0.089 | 0.042 | -0.012–0.163 |
| Lowest Hb (g/dl) | 0.115 | 0.111 | 0.107 | -0.078–0.304 |
| Early Hb (g/dl) | 0.048 | 0.050 | 0.146 | -0.198–0.330 |

* Regression coefficient retrieved from the generalized linear model with binomial family and logit link function (the numbers are rounded up to three decimal places)

CLCR: creatinine clearance, PCI: percutaneous coronary intervention, Hb: hemoglobin

We attempted to examine the topic from two perspectives to compare the current study's findings with other relevant research. First, we compared our findings to previous information regarding the higher involvement of the female sex in in-hospital mortality of STEMI patients to generate a complete picture. Next, we tried to compare the existing evidence on the effect of CLCR on in-hospital mortality in STEMI patients with our findings to obtain more precise conclusions. In addition, we will also briefly examine the role of other mediating variables studied. From the first standpoint, we first compared our findings with Siabani et al. study [1]. This study aimed to explain the sex differences in our study population. Although the results of this study at first glance support the findings of our study that women are not at higher risk of in-hospital death after PSM. However, due to the kind of statistical analysis used (regular regression analysis) and the number of variables analyzed, the study's results cannot be appropriately compared to the current study. In addition, the process of selecting the variables under examination is not described in Siabani et al. study [1], our study has a more extended period and, subsequently, a larger sample size than the mentioned study, and finally, we have clearly stated how to handle missing data, but it is not clear in the comparative study. In a similar study [15], PSM analysis was used to explain the differences between men and women with primary PCI. In the first 30 days after STEMI, even after matching, women experienced higher mortality rates, which is not consistent with the result of our study. However, in the follow-up, the two groups of survivors were the same. Perhaps this contradiction is due to selecting and managing related variables. Because in this research, variables with P-value less than 0.2 were selected for PSM, which is challenging, and mediating variables cannot be identified. Similarly, 3194 patients were included in multicenter research in India [16] between 2013 and 2017. When 510 pairs of patients were compared following PSM analysis, women had higher death rates throughout the one-year follow-up, which is inconsistent with the results of our study. However, the length of the follow-ups of the two studies is different. In another study conducted in Iran [22] between 2008 and 2013, conventional regression methods were employed to assess sex disparities in 1017 patients. The researchers stated in this study that after controlling for the confounding variables, women were no longer at a greater risk of in-hospital death than males, which is consistent with the results of our study. In this study, only 7 variables of age, HLP, diabetes, smoking, history of ischemic heart disease, and reperfusion therapy were included in the regression model; moreover, their variable selection approach was unclear. Recent meta-analysis result is also inconsistent with our result. However, the result of this study should be evaluated critically due to its low generalizability, only by searching the PubMed database, unclear eligibility criteria regarding studies with the PSM method, and not paying attention to the variable selected in the included studies and how they were selected [23]. By summarizing the points presented, it can be said that the enigma of sex differences in in-hospital mortality after STEMI has not yet been solved and remains strong.

Suppose we intend to look at the association between CLCR and in-hospital mortality in STEMI patients. As one of the first studies in this field, we can refer to the study of Santopinto et al. [24]. This study used data collected from 94 hospitals in 14 countries to assess the association between CLCR and two subsets of acute coronary syndromes (ACS), including STEMI. Finally, in this unique study, the independent role of CLCR on in-hospital mortality and patient bleeding was established. The results of this study provided the necessary basis for enhancing the management of cardiac patients with renal dysfunction. Although this study verifies our findings, it does not address sex differences or the role of CLCR as a mediator. Another study looked into the relationship between varying levels of creatinine and CLCR and in-hospital mortality and established a dose-response association between various levels of CLCR and in-hospital mortality in STEMI patients, where more severe levels of CLCR were related with increased mortality. Although the sex differences and, consequently, the

mediating role of CLCR was not mentioned in this study, it nevertheless supports the results of our study [25]. Another similar study reported the effect of CLCR, similar to the two previous studies discussed [26]. The association between chronic kidney disease (CKD) and short-term (in-hospital) and long-term mortality in STEMI patients was explored in another study [27], which can be stated to have the most confirming role on the results of our research. So that, the decrease in glomerular filtration rate (eGFR), based on CLCR, was reported as an influential independent variable in reducing mortality in STEMI patients. The study's most intriguing finding was that by including the eGFR term in the multiple regression model, the significant link between sex and outcome disappeared and women were no longer at a higher risk of mortality. Although the mediating effect is not addressed in this work, it coincidentally showed the mediating role of CLCR. Consequently, our work is the first or one of the first studies to reveal the mediating role of CLCR in the causal chain of sex and in-hospital death from an epidemiological standpoint.

As a secondary and clarifying result, even though most of our study participants were males, with around one woman for every 4–5 men, this is exceptionally usual, and past research has shown a similar pattern [16, 17, 28]. One explanation for this is that males have a higher occurrence of the illness at a younger age and, subsequently, a higher risk of disease. Women, on the other hand, are more vulnerable to heart disease as they become older [29, 30]. The complete explanation for this is beyond the scope of this research. Concerning other mediating variables, fortunately, there is good evidence that shows the validity of our results. In this regard, evidence has proven the influence of Hb on in-hospital mortality and one-year mortality [31, 32]. Similarly, there is similar evidence for PCI [33–35]. The role of diabetes has also been proven in reliable sources [36–39]. In general, it can be said that the selection of these variables as mediating variables is correct, and probably the studies that did not consider the potential influence of these factors have gone the wrong way.

## Limitations and strengths

Although this study was conducted in a tertiary and a reference center in the west of Iran, it may be argued that one of the limitations of the current study is that it is not a multicenter study to assess the external validity of findings, as a multicenter study depends on a larger population and provides more reliable results. However, the evidence presented in the preceding paragraphs seems sufficient to address this problem, as most of the themes support our findings. Another drawback that readers may consider for the present research is its non-randomized design, which does not seem to be completely reasonable based on the available information. Because, as far as we know, this is the first research to investigate the causal role of CLCR, and it may serve as the foundation for future randomized trials. Although clinical trial studies in this field are also very limited, the study by Jennifer et al. is one such study [40].

On the other hand, one of the current study's strengths is its relatively high accuracy in choosing variables before employing PSM analysis. As a result, the essential variables of PSM were selected based on past clinical information, suggestions, and methodological principles. It adequately explained the sex difference in STEMI patients' in-hospital mortality and identified a mediating association between CLCR and the causal chain between sex and in-hospital death, which can be clinically and medically relevant. As mentioned, previous evidence has supported our results in this regard. In other words, previous evidence reporting this relationship is anecdotal and unaware of its mediating role, which leaves no room for doubt. However, to complete the argument, we recommend that other researchers from around the world evaluate our results for probable correctness or inaccuracy. It is hoped that clinical management of these patients will soon change.

## Conclusions

Altogether, our study could help resolve sex difference in in-hospital mortality after STEMI. Sex no longer played an independent role in STEMI mortality after matching. In addition, the complete mediating role of CLCR in this causal chain was well elucidated, and CLCR alone could fully explain this relationship. Without exaggeration, this clinical finding can completely revolutionise existing knowledge in this field and remove ambiguities. As a result, adopting the proper epidemiological viewpoint can assist us in clarifying the network etiology of diseases.

## Supporting information

**S1 File. Patients data before propensity score matching.**
(RAR)

**S2 File. Supporting information—Including method clarification, data presentation, and Figs and Tables.**
(RAR)

**S3 File. All R and Stata codes to replicate study results.**
(RAR)

**S1 Graphical abstract.**
(TIF)

## Author Contributions

**Conceptualization:** Parisa Janjani, Nahid Salehi, Mohammad Rouzbahani, Soraya Siabani, Meysam Olfatifar.

**Data curation:** Parisa Janjani, Nahid Salehi, Mohammad Rouzbahani, Soraya Siabani, Meysam Olfatifar.

**Formal analysis:** Meysam Olfatifar.

**Funding acquisition:** Parisa Janjani.

**Investigation:** Parisa Janjani, Soraya Siabani.

**Methodology:** Meysam Olfatifar.

**Software:** Meysam Olfatifar.

**Supervision:** Parisa Janjani, Nahid Salehi, Mohammad Rouzbahani.

**Writing – original draft:** Meysam Olfatifar.

**Writing – review & editing:** Parisa Janjani, Nahid Salehi, Mohammad Rouzbahani, Soraya Siabani, Meysam Olfatifar.

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
