## [Decision Letter · Decision Letter 0]

20 Mar 2023

PONE-D-23-05629Creatinine clearance is key to solving the enigma of sex difference in in-hospital mortality after STEMI: propensity score matching and mediation analysisPLOS ONE

Dear Dr. Olfatifar,

Thank you for submitting your manuscript to PLOS ONE. After careful consideration, we feel that it has merit but does not fully meet PLOS ONE’s publication criteria as it currently stands. Therefore, we invite you to submit a revised version of the manuscript that addresses the points raised during the review process.

We look forward to receiving your revised manuscript.

Kind regards,

Hideo Kato

Academic Editor

PLOS ONE

Journal Requirements:

Reviewers' comments:

Reviewer's Responses to Questions

**Comments to the Author**

1. Is the manuscript technically sound, and do the data support the conclusions?

Reviewer #1: Partly

2. Has the statistical analysis been performed appropriately and rigorously? 

Reviewer #1: I Don't Know

3. Have the authors made all data underlying the findings in their manuscript fully available?

Reviewer #1: Yes

4. Is the manuscript presented in an intelligible fashion and written in standard English?

Reviewer #1: Yes

5. Review Comments to the Author

Reviewer #1: Major

Abstract

The abstract does not adequately summarise the content of the main text. In particular, the method section should be summarised accurately.

Introduction

The aim is to investigate the association between gender and hospital deaths due to STEMI. The statistical methods and other information are described, but not enough is said about the background to the analysis on mediating effects.

Discussion

The CLCR has not been examined for confounding with other influencing factors besides gender, such as weight and Scr values. This point should be mentioned.

minor

The first sentence on page 9 is duplicated.

Figure and Tables

Abbreviations or units are not mentioned. Vertical lines in the tables should be removed to make them easier to read.

Many of the abbreviations are not explained in the text either and should be reviewed in their entirety (such as LBBB, ECG in page 3).

6. PLOS authors have the option to publish the peer review history of their article (what does this mean?). If published, this will include your full peer review and any attached files.

Reviewer #1: No

---

## [Author Response · Author response to Decision Letter 0]

29 Mar 2023

Journal Requirements:

Dear editor, 

Thanks for your comments and suggestions, we tried to apply all your requests to the manuscript.

Answer:

Thanks, corrections were done.

Answer:

Thanks, corrections were done.

 

Reviewers' comments:

Reviewer #1: Major

Dear reviewer, 

Thanks for your comments and suggestions, we tried to apply all your requests in "Track-change" mode to the manuscript and highlighted all corrections.

Abstract

The abstract does not adequately summarise the content of the main text. In particular, the method section should be summarised accurately.

Answer: 

Thanks for your attention, corrections were done.

Introduction

The aim is to investigate the association between gender and hospital deaths due to STEMI. The statistical methods and other information are described, but not enough is said about the background to the analysis on mediating effects.

Answer: 

Thanks, corrections were done, we hope it will be satisfactory.

Discussion

The CLCR has not been examined for confounding with other influencing factors besides gender, such as weight and Scr values. This point should be mentioned.

Answer: 

Thanks so much, the aim of our research is to elucidate the causal chain between gender (exposure/treatment) and in-hospital mortality (outcome), based on all studied variables that CLCR is one of them, as we did not address this issue. So that, CLCR effect on the causal chain between sex and in-hospital mortality was evaluated together with other suspected intermediate variables. Therefore, drawing connections outside this domain is very difficult and ambiguous. Hence, we request from you waive this case if possible.

Minor

The first sentence on page 9 is duplicated.

Answer: Thanks a lot, the duplicate sentence was removed.

Figure and Tables

Abbreviations or units are not mentioned. Vertical lines in the tables should be removed to make them easier to read.

Answer:

Thanks, we added abbreviations and corresponding units for all variables in tables 1–3, where possible, plus appropriate abbreviations were added to figure legends, and vertical lines in tables were removed.

Many of the abbreviations are not explained in the text either and should be reviewed in their entirety (such as LBBB, ECG in page 3).

Answer:

Thanks for your attention, corrections were done, for each abbreviation, we used its full equivalent.

---

## [Decision Letter · Decision Letter 1]

5 Apr 2023

Creatinine clearance is key to solving the enigma of sex difference in in-hospital mortality after STEMI: Propensity score matching and mediation analysis

PONE-D-23-05629R1

Dear Dr. Olfatifar,

We’re pleased to inform you that your manuscript has been judged scientifically suitable for publication and will be formally accepted for publication once it meets all outstanding technical requirements.

Kind regards,

Hideo Kato

Academic Editor

PLOS ONE

Additional Editor Comments (optional):

Reviewers' comments:

Reviewer's Responses to Questions

**Comments to the Author**

1. If the authors have adequately addressed your comments raised in a previous round of review and you feel that this manuscript is now acceptable for publication, you may indicate that here to bypass the “Comments to the Author” section, enter your conflict of interest statement in the “Confidential to Editor” section, and submit your "Accept" recommendation.

Reviewer #1: All comments have been addressed

2. Is the manuscript technically sound, and do the data support the conclusions?

Reviewer #1: Yes

3. Has the statistical analysis been performed appropriately and rigorously? 

Reviewer #1: Yes

4. Have the authors made all data underlying the findings in their manuscript fully available?

Reviewer #1: Yes

5. Is the manuscript presented in an intelligible fashion and written in standard English?

Reviewer #1: Yes

6. Review Comments to the Author

Reviewer #1: The authors were appropriately addressed for the problems.

7. PLOS authors have the option to publish the peer review history of their article (what does this mean?). If published, this will include your full peer review and any attached files.

Reviewer #1: No
